# Antifungal Activity of Volatile Components from *Ceratocystis fimbriata* and Its Potential Biocontrol Mechanism on *Alternaria alternata* in Postharvest Cherry Tomato Fruit

Shijun Xing,[a] Yating Gao,[a] Xue Li,[a] Huan Ren,[a] Yang Gao,[a] Hui Yang,[a] Yanmei Liu,[a] Shuqi He,[a] Qiong Huang[a]

[a]State Key Laboratory for Conservation and Utilization of Bio-Resources in Yunnan, Yunnan Agricultural University, Kunming, People's Republic of China

**ABSTRACT** Infection by fungal pathogens is the main factor leading to postharvest rot and quality deterioration of fruit and vegetables. Rotting caused by *Alternaria alternata* is a concerning disease for numerous crops in both production and postharvest stages, especially tomato black spots. In this study, the double Petri dish assay showed that the VOCs of *Ceratocystis fimbriata* WJSK-1 and Mby inhibited the mycelial growth of fungal pathogen *A. alternata*, with a percentage inhibition of 52.2% and 42.9%. Then, HS-SPME-GC-MS technology was used to analyze the volatiles produced by two strains of *C. fimbriata* (WJSK-1, Mby), a total of 42 volatile single components were obtained, the main volatiles compounds identified include nine esters, 10 ketones, five alcohols, four aldehydes, three aromatic hydrocarbons, three heterocycles, four alkenes, three alkanes, and one acid. After that, the antifungal activity of a single volatile component was evaluated both *in vitro* and *in vivo*, four single components with antifungal effects were screened out, namely, benzaldehyde, nonanal, 2-Phenylethanol and isoamyl acetate, with $IC_{50}$ values show the smallest values for benzaldehyde and nonanal at 0.11 $\mu$L mL$^{-1}$, 0.04 $\mu$L mL$^{-1}$. *A. alternata* exposed to VOCs had abnormal morphology for hyphae, delayed sporulation, and inhibited spore germination. *In vivo* experiment shows that the four volatile components can effectively suppress disease incidence on fungal-inoculated fruit; the two aldehydes (benzaldehyde and nonanal) have more prominent effect on delaying fruit onset of disease. The results showed that VOCs produced by *C. fimbriata* have potential as a fumigant for controlling black rot in cherry tomatoes.

**IMPORTANCE** In this research, the volatile organic compounds (VOCs) produced based on *C. fimbriata* exhibited strong antifungal activity against the fungal pathogen *A. alternata*. Our aim is to explore their bacteriostatic components. HS-SPME-GC-MS technology was used to analyze the volatiles produced by the *C. fimbriata* strain (WJSK-1, Mby). Postharvest cherry tomato fruit black rot caused by *A. alternata* was treated both *in vitro* and *in vivo*, with pure individual components produced by *C. fimbriata*. The benzaldehyde, nonanal, 2-Phenylethanol, and isoamyl acetate from *C. fimbriata* can effectively inhibit growth of *A. alternata*, and delay disease. It has the potential to be developed as a new type of fumigant, a potential replacement for fungicides in the future.

**KEYWORDS** *Alternaria alternata*, volatile organic compounds (VOCs), antifungal activities, postharvest, tomato, fungal metabolites

Address correspondence to Shuqi He, 744994604@qq.com, or Qiong Huang, huangqiong2022@139.com.

Author order was determined on the basis of contributed to this work.

The authors declare no conflict of interest.

A s we know biocontrol agents (BCAs) as a biofumigant is a specific application in biological control since they are not in direct contact with the pathogen, microbial volatile organic compounds (VOCs) easily volatilize and degrade at room temperature. According to research, fungal VOCs have been developed as green chemicals and fuel sources known as "mycodiesel" (1). In this way, the use of VOCs for biological fumigants will be a promising work (2). Mainly reflected in several advantages of VOCs, on the one hand, they can diffuse in space, for example, under postharvest conditions,

VOCs can be easily used to control diseases of vegetables and fruit during storage or long-term transportation without the need for discharge from stores or containers. On the other hand, there is no direct contact between fruit and vegetables, antagonists to avoid residues, and drug damage (3, 4). Therefore, it is a promising application direction for VOCs to prevent postharvest decay as biological fumigation.

The occurrence of postharvest diseases is one of the main factors that cause economic losses during postharvest storage and transportation, which not only affects the storage time and shelf life of fruits and vegetables, but also leads to a decline in market value (5). Fresh vegetables and fruits are susceptible to pathogenic fungi from genera *Alternaria*, *Botrytis*, *Fusarium*, *Geotrichum*, *Penicillium*, and *Sclerotium*, under high temperature and humidity (6, 7). Postharvest cherry tomatoes (*Solanum lycopersicum*) are highly susceptible to some fungal pathogens, especially necrotrophic pathogens *A. alternata* (8, 9). *Alternaria alternata* can cause postharvest black rot in cherry tomatoes and a loss of their commodity value (6, 10). Moreover, mycotoxins produced by *A. alternata* have negative effects on human and animal health, which involves the occurrence of mutations, chromosomal aberrations, and DNA damage (11). Traditionally, postharvest decay control is achieved using chemical fungicides; however, concerns about the potential dangers of chemical synthetic fungicides to human health and the environment have prompted research into alternate management methods, such as biological control agents (12, 13). In addition to chemical synthetic fungicides, microorganisms that produce VOCs have recently received a lot of attention, VOCs are considered promising alternatives for postharvest fungi management (14, 15). A large number of studies have shown many antagonistic microorganisms, e.g., *Aureobasidium pullulans*, *Metschnikowia fructicola*, and *Bacillus amyloliquefaciens*, etc., can produce VOCs metabolites that can control the growth of postharvest fungal pathogens on fruit (16). *Bacillus subtilis* produced antagonistic volatile compounds that caused structural changes in six pathogenic fungi, including *Alternaria alternata*, *Cladosporium oxysporum*, *Fusarium oxysporum*, *Paecilomyces lilacinus*, *Paecilomyces variotii*, and *Pythium afertile* (17).

Related studies have confirmed that many microorganisms have been found to produce VOCs that have antifungal properties (18). In addition, most volatile organic compounds belong to five chemical groups: terpenes, fatty acid derivatives, benzene compounds, phenylpropanoid compounds, and amino acid derivatives (19). VOC detection and identification, such as solid-phase microextraction sample gas chromatography-mass spectrometry (SPME-GC-MS), has provided researchers with strong support in researching antibacterial VOCs of microorganisms (20). Therefore, the antibacterial mechanisms of bacteria and fungi capable of producing volatiles have been widely studied. Biocontrol applications for postharvest disease control are currently focusing on the use of biodegradable VOCs created by microorganisms; these kinds of volatile compounds exhibit strong antibacterial properties and pose no health risk (20, 21). More recently, the biocontrol ability of *Pichia anomala* (also known as *W. anomalous*) has been attributed to the production of 2-Phenylethanol, which inhibits *Aspergillus flavus* spore germination, growth, and toxin production. Volatile compound inhibition of aflatoxin B1 formation, in particular, was associated with a significant downregulation of clustering aflatoxin biosynthesis genes (22). Some microorganism VOCs components, such as dimethyl disulfide, dimethyl trisulfide, 3-hydroxy-2-butanone, and acetoin, have been identified as antifungal compounds (23–25). In addition, plant volatiles, including acetaldehyde, 2-E-hexenal, benzaldehyde, ethanol, acetic acid, essential oils (EOs), and microbial VOCs, have been identified as potential antimicrobial substances (15). Therefore, alternative strategies for reducing postharvest losses of vegetables and fruit caused by fungal pathogens are being extensively researched around the world (10).

*Ceratocystis fimbriata* is a typical soilborne ascomycete that initially attracted attention due to its ability to cause disease in a broad range of economically important plants. Since then, with extended research, it has been shown to be a new aroma-producing fungus, capable of producing pleasant volatile aromas, similar to the fruity

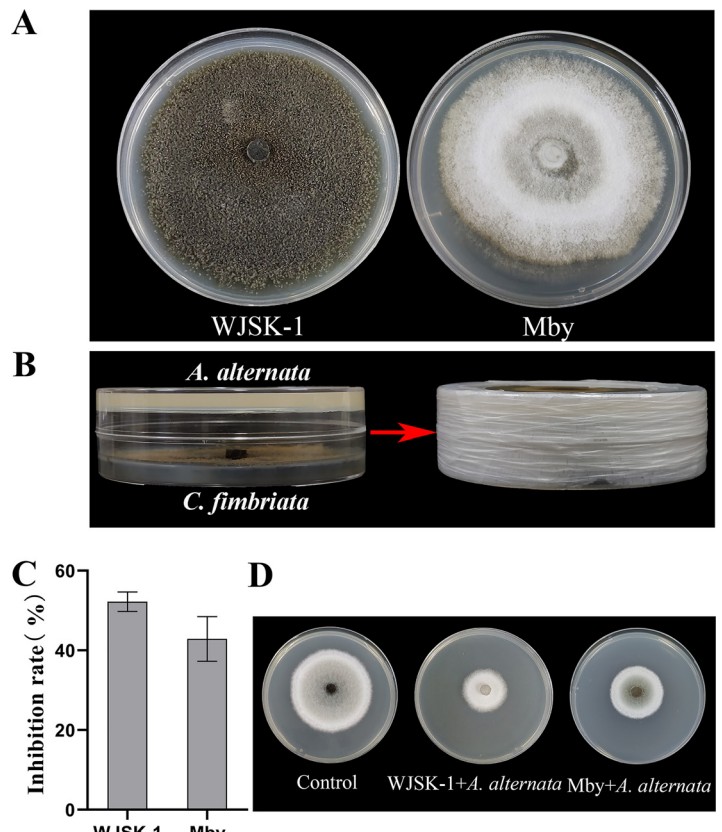

**FIG 1** The inhibitory effect of VOCs released from *C. fimbriata* strains (WJSK-1 and Mby) against fungal pathogen *A. alternata*, (A) Colony morphology of *C. fimbriata* strains, (B) double Petri Dish pair-button coculture, (C) percentage of inhibition, (D) flat plate inhibition effect.

aroma (26, 27). Based on the widely used GC-MS technique in the analysis of microbial volatiles, it has been reported that *C. fimbriata* is capable of producing a large number of VOCs, analysis revealed the presence at least 28 single VOCs from at least category five of organic compounds, such as acids, alcohols, esters, and lipids (28). Of interest is that it shows strong antagonistic effects against a variety of pathogenic fungi, including *Botrytis cinerea*, *Monilinia fructicola*, *Fusarium* sp., *Penicillium* sp., and *Rhizoctonia solani*. But the inhibition mechanism of single VOCs against fungus has not been investigated (27, 29). Therefore, it is meaningful to explore the volatiles from *C. fimbriata* bacteriostatic components and to reveal inhibition.

As a result, the objective of this study was to evaluate the antifungal activity of VOCs produced by *C. fimbriata* strains against *A. alternata* *in vivo* and *in vitro*, to explore its biocontrol potential in the prevention and control of tomato black spot, and to lay the foundation for the development of new fumigation agents.

## RESULTS

***C. fimbriata* strains WJSK-1 and Mby released VOCs inhibit the growth of *A. alternata*.** The test results show that WJSK-1 and Mby released VOCs with abilities for the inhibition of *A. alternata* growth (Fig. 1D). *A. alternata* growth in the tested plates was smaller than that in the control plates, and the colony morphology was changed. The percentage inhibitions of *A. alternata* by WJSK-1 and Mby was 52.2% and 42.9%, respectively (Fig. 1C), and the inhibitory effect of strain WJSK-1 is better than that of strain Mby.

**GC-MS analysis and screening of VOCs.** The VOCs produced by two *C. fimbriata* strains were analyzed using SPME-GC-MS. A total of 42 VOCs was identified from the strain WJSK-1 and Mby (Table 1). It mainly included nine esters, 10 ketones, five alcohols, four aldehydes, three Aromatic hydrocarbons, three heterocycles, four alkenes, three alkanes,

**TABLE 1** HS-SPME/GC-MS analysis of VOCs produced by *C. fimbriata* strains (WJSK-1 and Mby)

| No | Family | Identified compound[a] | CAS no. | Molecular formula | Retention time (min) | Area[b] (%) |
|---|---|---|---|---|---|---|
| 1 | Esters | Ethyl acetate | 141-78-6 | $C_4H_8O_2$ | 7.22 | 13.67 |
| 2 | | *n*-Propyl acetate | 109-60-4 | $C_5H_{10}O_2$ | 18.245 | 0.57 |
| 3 | | Isobutyl acetate | 110-19-0 | $C_6H_{12}O_2$ | 20.899 | 79.61 |
| 4 | | Butyl acetate | 123-86-4 | $C_6H_{12}O_2$ | 22.604 | 0.02 |
| 5 | | Isobutyl propionate | 540-42-1 | $C_7H_{14}O_2$ | 24.707 | 0.13 |
| 6 | | Isoamyl acetate | 123-92-2 | $C_7H_{14}O_2$ | 25.151 | 7.58 |
| 7 | | 2-Methylbutyl acetate | 624-41-9 | $C_7H_{14}O_2$ | 25.274 | 3.48 |
| 8 | | 4-Pentenyl Acetate | 1576-85-8 | $C_7H_{12}O_2$ | 25.603 | 0.05 |
| 9 | | Benzyl acetate | 140-11-4 | $C_9H_{10}O_2$ | 36.508 | 0.06 |
| 10 | Ketones | Acetone | 67-64-1 | $C_3H_6O$ | 9.274 | 0.02 |
| 11 | | 2-Butanone | 78-93-3 | $C_4H_8O$ | 23.342 | 10.18 |
| 12 | | 2-Acetoxyl-3-butanone | 4906-24-5 | $C_6H_{10}O_3$ | 27.234 | 2.88 |
| 13 | | Acetoin | 513-86-0 | $C_4H_8O_2$ | 20.172 | 0.17 |
| 14 | | 6-methyl-3-Heptanone | 624-42-0 | $C_8H_{16}O$ | 29.982 | 0.01 |
| 15 | | 2-Heptanone | 110-43-0 | $C_7H_{14}O$ | 21.214 | 0.86 |
| 16 | | 2-Undecanone | 112-12-9 | $C_{11}H_{22}O$ | 19.49 | 0.54 |
| 17 | | 2-Tridecanone | 593-08-8 | $C_{13}H_{26}O$ | 24.479 | 0.57 |
| 18 | | 2-Tetradecanone | 2345-27-9 | $C_{14}H_{28}O$ | 25.804 | 0.74 |
| 19 | | 2-Pentadecanone | 2345-28-0 | $C_{15}H_{30}O$ | 28.991 | 0.54 |
| 20 | Alcohols | 2-Phenylethanol | 60-12-8 | $C_8H_{10}O$ | 35.916 | 0.01 |
| 21 | | 3,3-Dimethylbutane-2-ol | 464-07-3 | $C_6H_{14}O$ | 34.267 | 0.05 |
| 22 | | Citronellol | 106-22-9 | $C_{10}H_{20}O$ | 17.932 | 3.37 |
| 23 | | 1-Octen-3-ol | 3391-86-4 | $C_8H_{16}O$ | 10.03 | 0.53 |
| 24 | | 2-Tert-Butyl-6-methylphenol | 2219-82-1 | $C_{11}H_{16}O$ | 22.557 | 2.14 |
| 25 | Aldehydes | Benzaldehyde | 100-52-7 | $C_7H_6O$ | 30.244 | 0.05 |
| 26 | | Butanal | 123-72-8 | $C_4H_8O$ | 1.933 | 0.70 |
| 27 | | Hexanal | 66-25-1 | $C_6H_{12}O$ | 5.502 | 8.76 |
| 28 | | Nonanal | 124-19-6 | $C_9H_{18}O$ | 14.261 | 2.05 |
| 29 | Aromatic hydrocarbons | Ethylbenzene | 100-41-4 | $C_8H_{10}$ | 24.989 | 0.05 |
| 30 | | 1,3-Dimethyl-Benzene | 108-38-3 | $C_8H_{10}$ | 7.137 | 12.34 |
| 31 | | p-Xylene | 106-42-3 | $C_8H_{10}$ | 8.089 | 2.76 |
| 32 | Heterocycles | Ethylene oxide | 75-21-8 | $C_2H_4O$ | 6.515 | 0.02 |
| 33 | | 2-Pentyl-Furan | 3777-69-3 | $C_9H_{14}O$ | 10.219 | 1.72 |
| 34 | | Formamide | 75-12-7 | $CH_3NO$ | 8.108 | 0.07 |
| 35 | Alkenes | Styrene | 100-42-5 | $C_8H_8$ | 26.214 | 0.49 |
| 36 | | Limonene | 138-86-3 | $C_{10}H_{16}$ | 30.642 | 0.45 |
| 37 | | (S)- (-) Limonene | 5989-54-8 | $C_{10}H_{16}$ | 30.642 | 0.05 |
| 38 | | Bicyclohepta-2,6-diene | 2422-86-8 | $C_7H_8$ | 20.588 | 0.02 |
| 39 | Alkanes | Isobutane | 75-28-5 | $C_4H_{10}$ | 15.414 | 0.05 |
| 40 | | Tridecane | 629-50-5 | $C_{13}H_{28}$ | 25.894 | 0.21 |
| 41 | | 3-Methyl-5-propylnonane | 31081-18-2 | $C_{13}H_{28}$ | 26.425 | 2.35 |
| 42 | Acid | Acetic acid | 64-19-7 | $C_2H_4O_2$ | 15.561 | 0.04 |

[a]Sum of single compounds detected in VOCs produced by two strains.
[b]Relative area of single compound as a percentage of the total area of the chromatographic peaks.

and one acid. And the proportion of ketones is the highest, with 10 ketone compounds, and acid compounds are the least.

**VOCs inhibited the mycelial radial growth of *A. alternata*.** To know whether VOCs of *C. fimbriata* had an effect on *A. alternata* in the postharvest cherry tomato control, a biological fumigation method was established to evaluate the antifungal activity of a single volatile component of *C. fimbriata in vitro*. The results show that VOCs treatment inhibited *A. alternata* mycelial radial growth, and optical microscopy results show that the morphology of the target fungus hyphae has changed. The four pure VOCs inhibited the growth of *A. alternata* mycelia, with some differences between the four VOCs (Table 2). Among the four synthetic pure VOCs, only nonanal completely inhibited the mycelial growth of *A. alternata* with concentrations of 0.1 $\mu$L mL$^{-1}$. Meanwhile, benzaldehyde and 2-Phenylethanol could both 100% inhibit mycelial radial growth when their final concentrations reached 0.6 $\mu$L mL$^{-1}$, IC$_{50}$ is 0.11 $\mu$L mL$^{-1}$, 0.18 $\mu$L mL$^{-1}$, respectively; however, at 0.6 $\mu$L mL$^{-1}$, isoamyl acetate inhibited the mycelial growth of *A. alternata* by 81% (Fig. 2A and B). The nonanal showed the best antifungal activity with the least IC$_{50}$ was 0.04 $\mu$L mL$^{-1}$.

**TABLE 2** Effects of tested compounds on the growth of *A. alternata* mycelial growth after incubation at 25°C for 7 days[a]

| Volatile compound | Mycelial growth inhibition rate (%) | | | | | | IC$_{50}$ $\mu$L mL$^{-1}$ |
| --- | --- | --- | --- | --- | --- | --- | --- |
| | 0.1 $\mu$L mL$^{-1}$ | 0.2 $\mu$L mL$^{-1}$ | 0.3 $\mu$L mL$^{-1}$ | 0.4 $\mu$L mL$^{-1}$ | 0.5 $\mu$L mL$^{-1}$ | 0.6 $\mu$L mL$^{-1}$ | |
| Benzaldehyde | 41.3 ± 14.2b | 100a | 100a | 100a | 100a | 100a | 0.11 |
| Nonanal | 100a | 100a | 100a | 100a | 100a | 100a | 0.04 |
| 2-Phenylethanol | 4.7 ± 4.5d | 65.3 ± 5.1c | 80.7 ± 3.0b | 92.3 ± 6.0a | 100a | 100a | 0.18 |
| Isoamyl acetate | 3.0 ± 1.7d | 28.0 ± 2.6c | 28.7 ± 3.2c | 30.3 ± 4.6c | 57.7 ± 5.5b | 81.3 ± 2.3a | 0.43 |

[a]Each value is the mean of three replicates ± standard deviation. The same lowercase letters indicate no significant difference between treatments ($P < 0.05$).

**Effect of single compounds on the mycelium morphology of *A. alternata*.** Microscopic analysis of *A. alternata* treated with volatile compounds, as well as micrographs of *A. alternata* hyphae in the untreated control, control group revealed a normal morphology with a smooth surface and uninflated and unstalked nodes (Fig. 3), and untreated samples had abundant mycelial vigorous growth. In contrast, the fungal hyphae exposed to volatiles experienced severe deformation (Fig. 3A) and twist (Fig. 3B), and some of the mycelia showed stalk nodes and expansion in the middle of the mycelium (Fig. 3D). Mycelial growth was suppressed and terminal hyphae showed stunted tips compared to the untreated control (Fig. 3C) (24). Another interesting phenomenon was that *A. alternata* exposed to volatiles appeared to delay sporulation, or no spores were observed. The control, in contrast, sporulated (Fig. 3).

**Effect of single compounds on spore germination of *A. alternata*.** The effects of different concentrations of the single compounds on the spore germination of *A. alternata* was tested *in vitro* (Fig. 4). The results indicated that compounds treatment inhibited spore germination of *A. alternata* compared with the control after 12 h of incubation at 25°C. Four single compounds did not completely inhibit spore germination at a concentration of *100 $\mu$L* mL$^{-1}$, with the spore germination rate decrease ranging from 60.9% to 74.7%. Spore germination was completely inhibited when the compound concentration increased to *200 $\mu$L* mL$^{-1}$ (Table 3).

**Evaluation of antifungal activity *in vivo*.** The effect of VOCs on the control of the black spots in tomato fruit was investigated. The total volatiles and 21 pure volatiles were tested against postharvest fungal in tomato *A. alternata*. After artificial inoculation with the pathogen, the four volatile compounds, i.e., benzaldehyde, nonanal, 2-Phenylethanol, and isoamyl acetate, were found to have a delayed pathogen diffusion effect on tomato fruit, and measurements of fruit disease spot diameter revealed differences between the compounds (Table 4). The disease inhibition of tomato fruit black spot was completely inhibited by selected four VOCs at a concentration of 0.2 $\mu$L mL$^{-1}$ (Fig. 5B-a). As shown in Fig. 4B, benzaldehyde exerted the strongest inhibitory effects, with only 0.05 $\mu$L mL$^{-1}$ achieving complete inhibition of disease decay. Followed by nonanal and 2-Phenylethanol (Fig. 5B-a and B-c), isoamyl acetate had the worst effect, with 0.2 $\mu$L mL$^{-1}$ not completely inhibit disease decay (Fig. 5B-d).

However, the disease incidence and lesion diameter of cherry tomato fruit after wound inoculation were also downtrend following the biological fumigation by four VOCs (Fig. 5A). The findings revealed that biological fumigation of volatile compounds had a clear effect on the control of tomato fruit black spots.

## DISCUSSION

Postharvest decay is one of the main factors that determine losses and food quality of vegetables and fruit. One of the most harmful fungi, *A. alternata*, can cause postharvest rot and black rot in tomatoes after harvest (30). As the disadvantages (toxicity and generation of pathogen resistance) of chemically synthesized fungicides gradually emerge, microbial metabolites will become one of the most likely replacements. Microorganisms produce a variety of VOCs, which are gas-phase, carbon-based compounds that can diffuse through the atmosphere and soils due to their small size (14). Therefore, the use of volatiles naturally produced by microorganisms for "biological fumigation" has become one of the research hot spots (15, 31). Inhibition of the

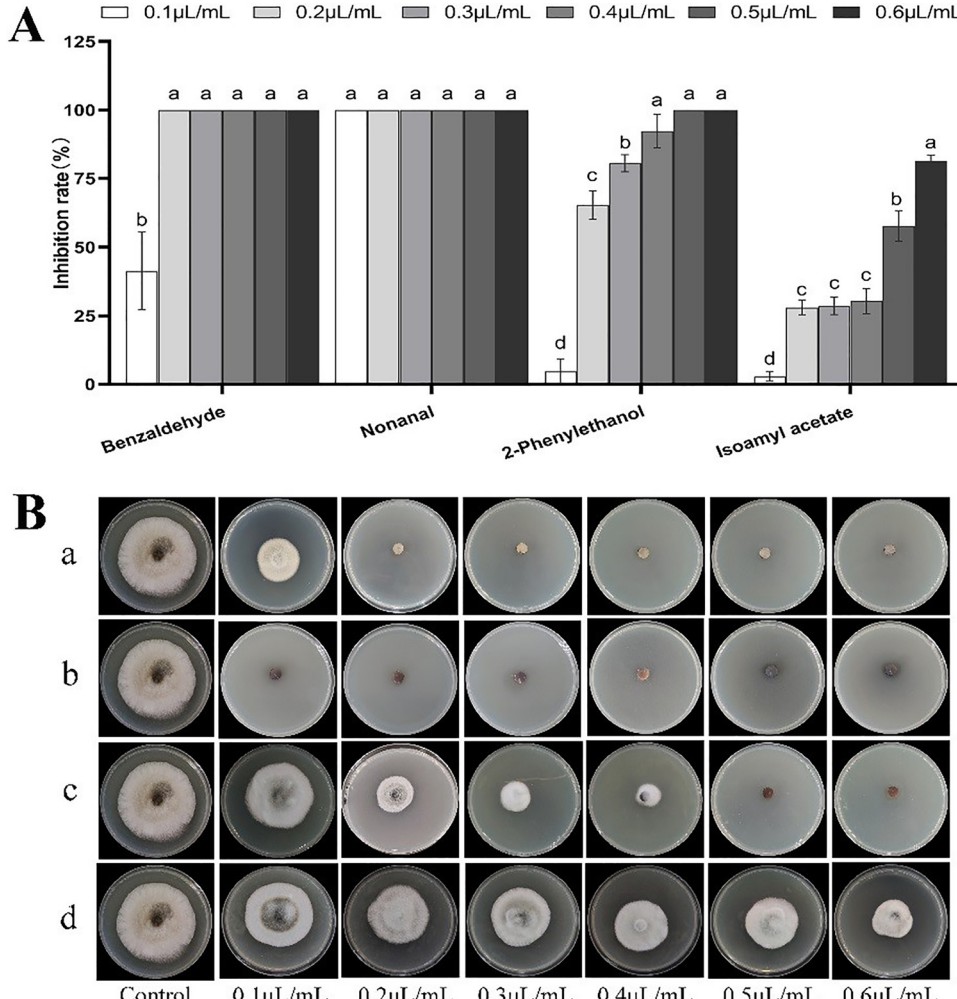

**FIG 2** Antifungal activity of the VOCs from *C. fimbriata* against *A. alternata*. After the four volatile components were fumigated for 7 days, the average diameter of colonies was measured. (A) Each bar represents the mean of three replicates of value ± standard. The same lowercase letters indicate no significant difference between treatments at the $P < 0.05$ level according to Duncan's multiple range test. (B) Plate antifungal activities of four compounds against mycelial growth of *A. alternata* after 7 days incubation at 25°C, (B-a) benzaldehyde; (B-b) nonanal; (B-c) 2-Phenylethanol; (B-d) isoamyl acetate.

growth of plant pathogens by VOCs derived from microorganisms has gained widespread attention (13). According to previous research, volatile-producing fungus, bacteria, and plant have a positive effect on biological fumigation for the prevention and control of fruit postharvest diseases. Fungi such as *Muscodor albus*, *Penicillium* (*Penicillium expansum*, *Penicillium italicum*), *C. fimbriata*, endophytic fungi (27, 32–34), bacteria such as *Bacillus velezensis*, *Bacillus siamensis*, *Pseudomonas fluorescens*, *Corallococcus. sp.* (31, 35, 36), and the plant *Origanum vulgare* (37), and their volatiles can effectively inhibit the growth of many pathogens, including *Botrytis cinerea*, *Colletotrichum*, *Geotrichum*, *Monilinia*, *Penicillium*, *Rhizopus*, *Macrophomina phaseolina*, *Monilinia fructicola*, and *Fusarium* (Table 5). As shown in this study, the VOCs produced by *Aureobasidium pullulansalso* also inhibit the growth of *A. alternata* (Table 5). In addition, some plant products have been recommended as safe alternatives to synthetic antimicrobials, e.g., Essential oils (9, 37, 38).

It has been previously reported that VOCs from *C. fimbriata* had a strong inhibitory effect on the test fungi, oomycetes, and bacteria, but the inhibition mechanism of single VOCs against *A. alternata* had not been investigated (27). With the exception that 2-Phenylethanol from *Aureobasidium pullulans* inhibited *A. alternata* (Table 5), this study

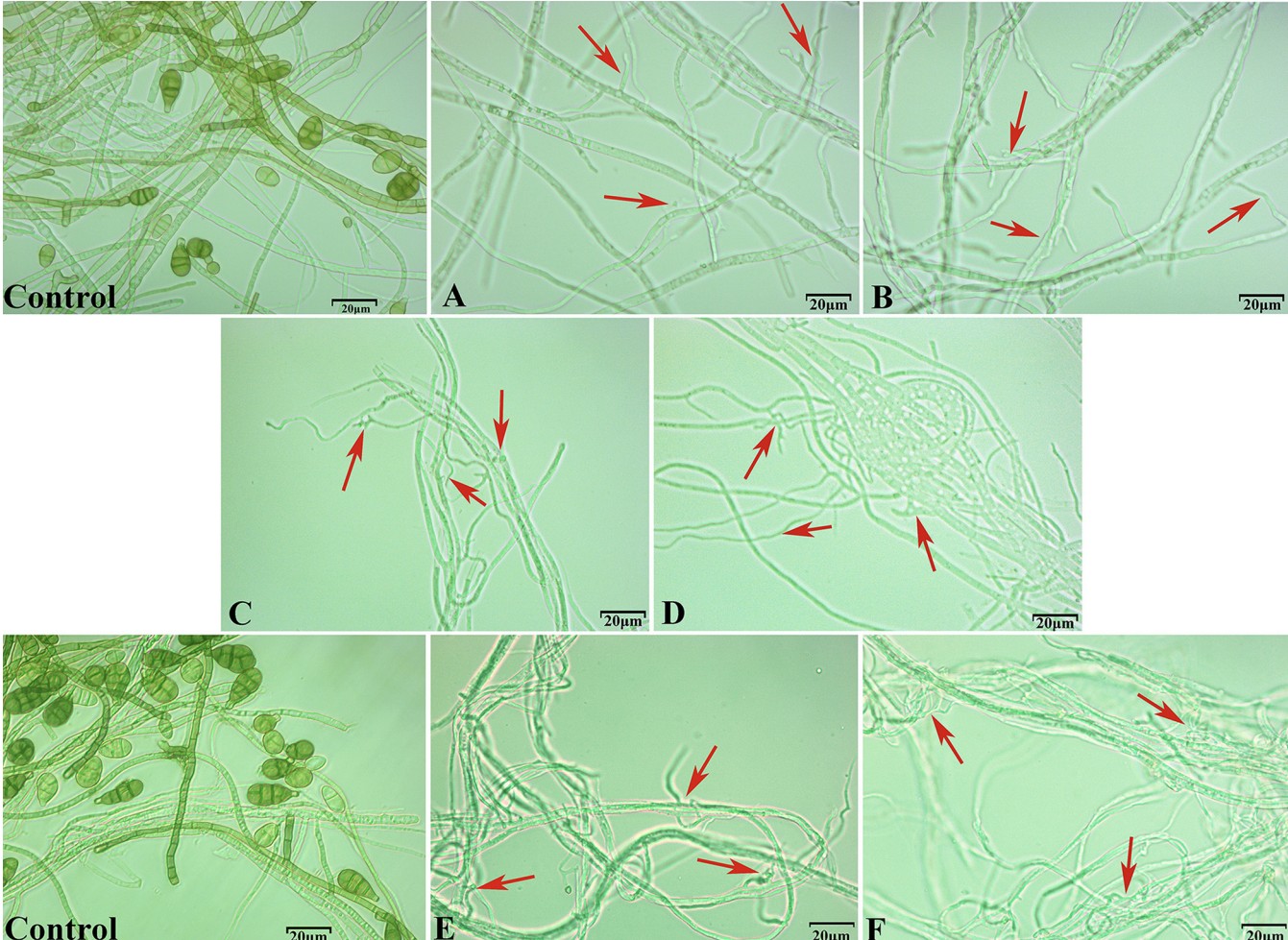

**FIG 3** Effect of VOCs from *C. fimbriata* on the hyphal morphology of *A. alternata*. After exposure to the four pure volatiles and the total VOCs of *C. fimbriata*, and incubation on potato dextrose agar (PDA) at 25°C for 7 days, microscopic morphological changes of *A. alternata*, (A) benzaldehyde, (B) nonanal, (C) 2-Phenylethanol, (D) isoamyl acetate, (E) strain WJSK-1, (F) strain Mby.

found that other three single volatile components from *C. fimbriata* can inhibit the growth of *A. alternata*, i.e., benzaldehyde, nonanal, and isoamyl acetate. In addition, Phenylethyl alcohol from endophytic fungi inhibited *Fusarium*, benzaldehyde, and 1-Phenylethanol, and nonanal from *B. velezensis* inhibited *B. cinerea* and *Penicillium* (Table 5). Based on Table 5, benzaldehyde, nonanal, and 2-Phenylethanol showed antifungal activity, and isoamyl acetate was reported to be effective in inhibiting *Curvularia lunata* (39). Furthermore, 2-Phenylethanol is not only a common ingredient in flavors (20), but also has excellent antifungal activity (40).

At present, the antifungal mechanism of microbial volatiles has been widely studied (Table 5), and mechanisms of action of VOCs as antimicrobial agents include the negative effects on fungal physiological functions (41). Pathogenic fungi mycelia of exhibit swelling and dehydration deformities, protoplast aggregation, and mitochondrial enlargement (42). The increase in ROS in hyphae cells structural defects multivesicular structures disruption, increases cell wall degrading enzymes, and chitinases (43, 44). Disruption of cell membrane integrity leads to leakage of cellular components and oxidative stress (45). Both VOCs were produced by microorganism and their single compound. The main performance is inhibited spore germination, and spores were deformed, the hyphae ruptured, shrunken, and twisted (20, 46, 47). Our findings are consistent with the above. We speculate the four compounds caused physiological damage to *A. alternata* hyphae and conidia, resulting in the decline in growth rate. In our study, *A. alternata* hyphae exposed to 2-Phenylethanol,

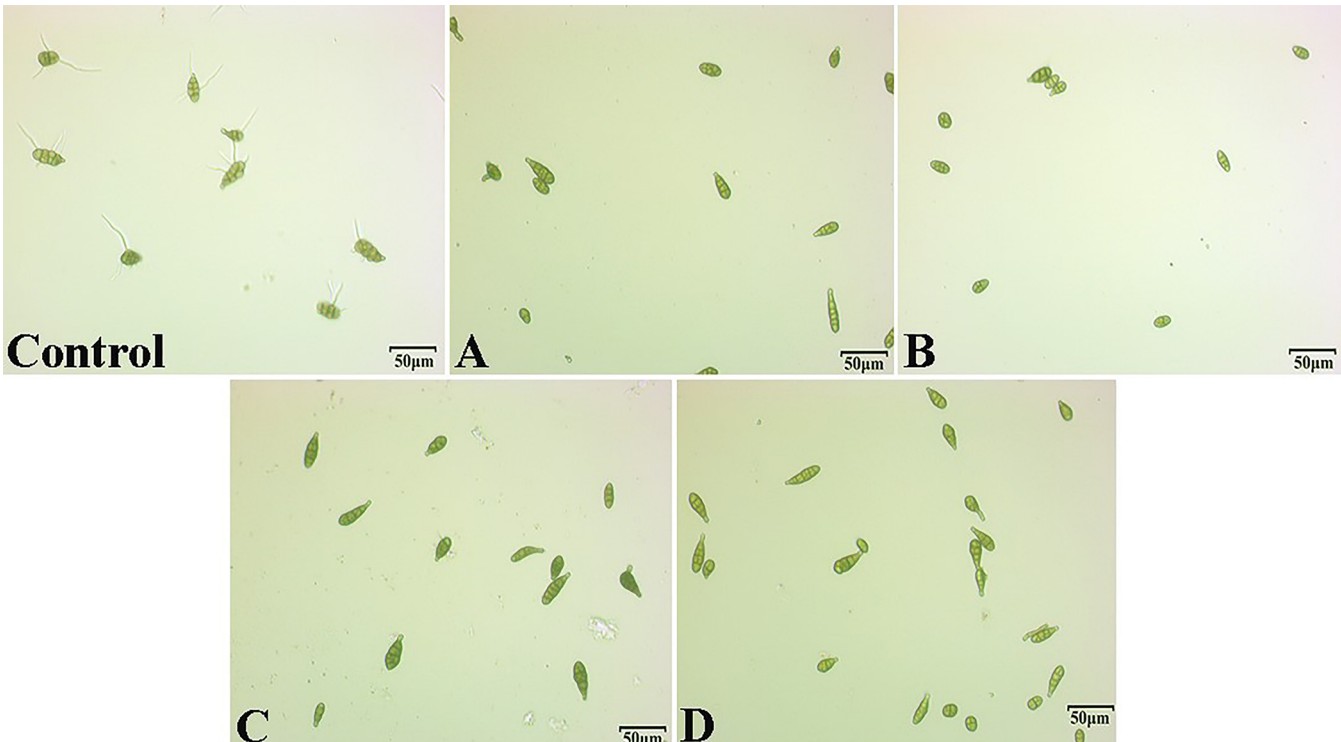

FIG 4 Determination of spore germination of *A. alternata* treated with different compounds at the same concentration (200 $\mu L$ mL$^{-1}$) after 12 h, (A) benzaldehyde, (B) nonanal, (C) 2-Phenylethanol, (D) isoamyl acetate.

benzaldehyde, nonanal, and isoamyl acetate showed dramatic morphological alterations (twist, knot, swell), and there were no spores in the visual field of the treatment group, while many spores could be seen in the control group (Fig. 3). Four of the single compounds can effectively inhibit the spore germination of *A. alternata* (Fig. 4). The IC$_{50}$ of nonanal and benzaldehyde were 0.04 $\mu L$ mL$^{-1}$ and 0.11 $\mu L$ mL$^{-1}$.

The antifungal activity of microbial VOCs may be due to the synergistic effect of all VOCs or the effect of a single component. Some single components produced by fungi and bacteria have been shown to have antifungal effects against some major pathogens (*B. cinerea*, *A. alternata*, *Penicillium*, *Fusarium*) (Table 5). Among the single components that have been reported to have antifungal activity in a part listed in Table 5, three components, namely, benzaldehyde, nonanal, and 2-Phenylethanol, were also detected in the VOCs of the *C. fimbriata* strains, and their antifungal activities were confirmed again in our study. There have been few studies on antifungal activity volatiles from *C. fimbriata*. To our knowledge, this is the first report of antifungal activity of a single component from *C. fimbriata* against *A. alternata*.

In conclusion, this study assessed an antifungal mechanism based on the volatile of *C. fimbriata*, which has been especially active against *A. alternata*. The findings indicate that its volatile components, 2-Phenylethanol, benzaldehyde, nonanal, and isoamyl

**TABLE 3** Effect of four compounds on *A. alternata* spore germination[a]

| Volatile compound | Conidia germination rate (%) | | |
| --- | --- | --- | --- |
| | 50 $\mu L$ mL$^{-1}$ | 100 $\mu L$ mL$^{-1}$ | 200 $\mu L$ mL$^{-1}$ |
| Benzaldehyde | 10.0 ± 2.6b | 7.0 ± 2.5b | 0.0 ± 0.0c |
| Nonanal | 8.1 ± 5.8b | 5.5 ± 2.3bc | 0.0 ± 0.0c |
| 2-Phenylethanol | 14.8 ± 3.3b | 7.8 ± 4.9b | 0.0 ± 0.0c |
| isoamyl acetate | 21.9 ± 3.2b | 8.8 ± 2.7c | 0.0 ± 0.0d |
| Control | 82.8 ± 4.0a | | |

[a]Values are mean ± standard deviation. The same lowercase letters indicate no significant difference between treatments ($P < 0.05$).

**TABLE 4** Effect of fumigation of four pure VOCs from *C. fimbriata* on disease spot diameter of cherry tomato after incubation at 25°C for 7 days[a]

| Volatile compound | Lesion diam (mm) | | | | | |
|---|---|---|---|---|---|---|
| | Control | 0.0125 $\mu L$ mL$^{-1}$ | 0.025 $\mu L$ mL$^{-1}$ | 0.05 $\mu L$ mL$^{-1}$ | 0.1 $\mu L$ mL$^{-1}$ | 0.2 $\mu L$ mL$^{-1}$ |
| Benzaldehyde | 30.7 ± 3.3a | 18.0 ± 3.5b | 15.0 ± 1.9c | 0.0 ± 0.0 d | 0.0d | 0.0d |
| Nonanal | 30.7 ± 3.3a | 29.9 ± 2.5a | 20.6 ± 6.1b | 2.6 ± 5.7c | 2.6 ± 5.7c | 0.0c |
| 2-Phenylethanol | 30.7 ± 3.3a | 28.7 ± 1.6a | 27.0 ± 1.6a | 16.8 ± 6.5b | 6.7 ± 7.1c | 0.0d |
| Isoamyl acetate | 30.7 ± 3.3a | 30.2 ± 2.4a | 24.3 ± 2.1b | 15.8 ± 2.6c | 15.5 ± 1.9c | 7.4 ± 0.4d |

[a]Each value is the mean of 10 fruit ± standard error. According to Duncan's ANOVA test at the $P < 0.05$ level, the same lowercase letters indicate no significant difference between treatments.

acetate, can inhibit *A. alternata* growth *in vitro* and *in vivo*. This study enriches our knowledge of VOCs from *C. fimbriata*, and these antifungal activity substances exhibit potential value as fumigants to control postharvest diseases.

## MATERIALS AND METHODS

**Plant material.** The cherry tomatoes used in this study were purchased from a local market in Kunming. The cherry tomato fruit used in this study did not receive any postharvest fungicide, and treatment tomato fruit with similar color, size, weight, and maturity were selected and randomly divided into 10 tomato fruit for each treatment. Before the experiment, tomato fruit was soaked in 75% ethanol for 30 s and naturally dried in the ultra-clean workstation. After that, surface wounds were scratched (round wounds 5 mm in diameter) along the equators.

**Fungal pathogen and antagonistic strains.** The fungal pathogen strain was isolated from postharvest tomato fruit and kept on potato dextrose agar (PDA) in the dark at 25°C. The *C. fimbriata* strains WJSK-1 (isolated from *Lactuca sativa var angustana Irish*) and Mby (isolated from *Musa basjoo Sieboid*) were identified. rRNA gene sequences for the two strains can be found in GenBank, accession numbers MH535912 and KY580883. Before the experiment, the fungal pathogen was grown and maintained on PDA at 25°C and purified twice for backup. The colony of strains WJSK-1 and Mby were shown Fig. 1A.

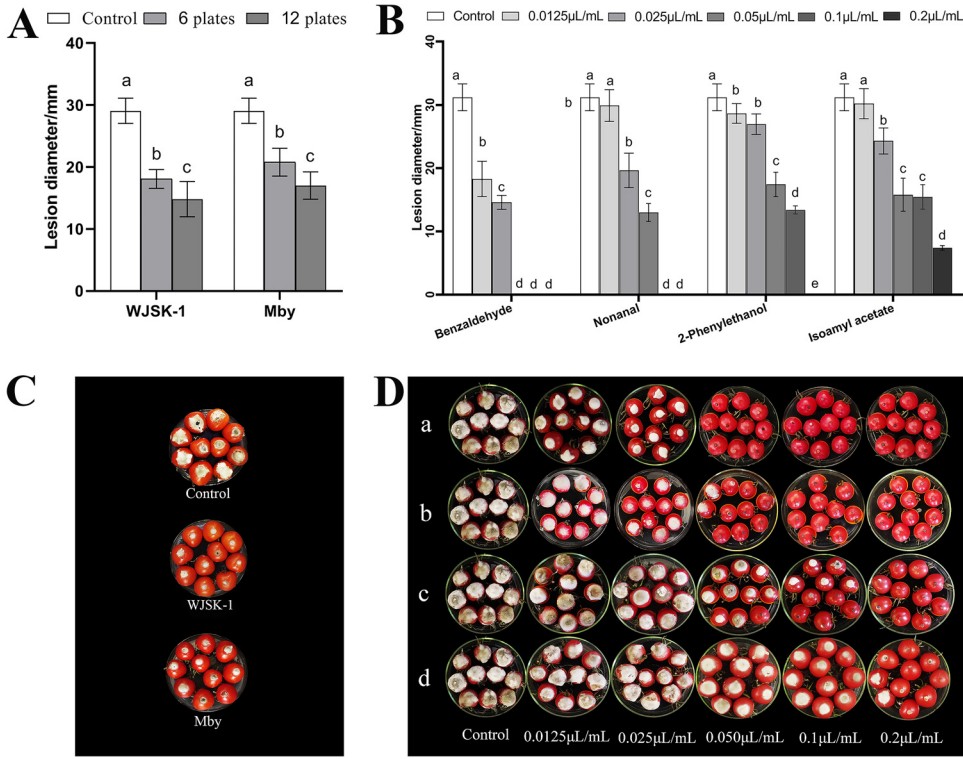

**FIG 5** Antifungal effect of *C. fimbriata* volatiles and four single VOCs *in vivo*. Inhibitory effect activities of *C. fimbriata* volatiles and four single compounds against *A. alternata* on cherry tomatoes after 7 days of storage at 25°C. (C) Treatment with *C. fimbriata* volatiles. Treatment with (D-a) benzaldehyde; (D-b) nonanal; (D-c) 2-Phenylethanol; (D-d) isoamyl acetate; (A and B) lesion diameter of different tomato fruit treated with *C. fimbriata* volatiles, and four single components. Different letters indicate differences ($P < 0.05$) in the same storage time and the same fungus from different treatments. Bars represent mean values ± standard deviation of 10 fruit.

**TABLE 5** Pure volatiles ingredients from microorganisms against fungal postharvest pathogens

| Fungal pathogens | Antagonistic microorganisms | Major antifungal compounds | Inhibitory effect | | Reference |
|---|---|---|---|---|---|
| | | | MIC | IC$_{50}$ | |
| *B. cinerea* | *Bacillus velezensis* | 1-Phenylethanol | 0.938 mL L$^{-1}$ | | 31 |
| | | Benzaldehyde | 0.062 mL L$^{-1}$ | | |
| | | Diacetyl | 0.006 mL L$^{-1}$ | | |
| | *Origanum vulgare* | Thymol carvacrol | 7.81 mg L$^{-1}$ | | 37 |
| *Alternata* | *Bacillus siamensis* LZ88 | 2-methylbutanoic acid | | 83.10 mg mL$^{-1}$ | 46 |
| | | 3-methylbutanoic acid | | 104.19 mg mL$^{-1}$ | |
| | | Citronella oil | 1 $\mu$L mL$^{-1}$ | | 6 |
| | | 2-Phenylethyllsothiocyanate | 1.22 mM | | 52 |
| | *Aureobasidium pullulans* | Ethanol | 524 mg L$^{-1}$ | | 20 |
| | | 2-Phenylethanol | 3.6 mg L$^{-1}$ | | |
| | | Laurel nobilus oil | 800 $\mu$g mL$^{-1}$ | | 53, 54 |
| | | Cassia oil | 300 ppm | | 9 |
| | | Thyme Oil | 500 ppm | | |
| *Penicillium.* | | $\alpha$-Phellandrene | 1.7 mL L$^{-1}$ | | 54 |
| | | Nonanal | 0.3 mL L$^{-1}$ | | |
| | *Bacillus velezensis* | Benzaldehyde | 0.125 mL L$^{-1}$ | | 31 |
| | | Diacetyl | 0.025 mL L$^{-1}$ | | |
| | | 1-Butanol | 0.150 mL L$^{-1}$ | | |
| | *Pseudomonas fluorescens* ZX | Dimethyl disulfide | 100 $\mu$L L$^{-1}$ | | 35 |
| | | Dimethyl trisulfide | 10 $\mu$L L$^{-1}$ | | |
| | *Corallococcus. sp.* EGB | Isooctanol | 4.0 $\mu$L/plate | | 36 |
| *Fusarium.* | Endophytic fungi | Phenylethyl alcohol, | | >1,000 $\mu$g mL$^{-1}$ | 55 |
| | | 2-methyl-1-butanol | | | |
| | | 3-methyl-1-butanol | | | |
| | | Monoterpenes eucalyptol | | | |
| | | Ocimene terpinolene | | | |
| | *Corallococcus. sp.* EGB | Isooctanol | 3.75 $\mu$L/plate | | 36 |

**Inhibitory effect of VOCs from strains WJSK-1 and Mby on *A. alternata*.** Our aim was to study the inhibitory effect of the *C. fimbriata* strains on the fungal pathogen *A. alternata*. Hence, we investigated the antifungal activities of VOCs from the *C. fimbriata* strains on mycelial growth of *A. alternata* by the double Petri dish assay (Fig. 1B) (48). In short, *C. fimbriata* strains were grown on PDA for 7 days The lid of each Petri dish was removed, and the plate containing PDA inoculated with tested fungi was replaced, and the two plates were then sealed together with Parafilm. The dual-culture plates were then incubated at an ambient temperature (25°C) for 7 days. PDA plates not inoculated with *C. fimbriata* strain served as a control. The colony diameter of *A. alternata* in the control and treatment groups were measured, and the colony morphology observed. All the experiments were repeated three times, and the percentage of inhibitions was calculated as follows:

$$\text{Inhibition of mycelial growth} \, (\%) \; = \; [(R_1 - R_2)/R_1] \; \times \; 100$$

where R$_1$ represents the diameter of the *A. alternata* in control and R$_2$ represents the diameter of the *A. alternata* with treatment.

**GC-MS analysis of *C. fimbriata* and preliminary screening of antimicrobial components. (i) GC-MS analysis of VOCs by *C. fimbriata*.** The SPME fiber was DVB/CAR/PDMS desorbed into the headspace VOCs of the double Petri-dish system and equilibrated at room temperature for 30 min. Injected into a DB-Wax column (with 30 m length, 250 $\mu$m inner diameter, and 0.25 $\mu$m thickness; Agilent, USA) with helium at a flow rate of 1 mL min$^{-1}$. The oven temperature was set to 40°C for 5 min before ramping from 40°C to 200°C at 5°C for 1 min, 200°C to 240°C at 10°C for 1 min, and holding for 5 min. The mass spectrometer was used to scan a mass range of 33 *m/z* to 350 *m/z* using electron impact (EI) ionization at 70 eV and an iron source temperature of 230°C. The GC-MS results were compared to the spectral database of the National Institute of Standards and Technology (NIST) (USA).

**(ii) Pure VOCs of antimicrobial components.** Twenty-one compounds were selected from Table 1 and the antifungal activity of single volatile compounds were determined both *in vitro* and *in vivo* for pathogen *A. alternata*. The volatile compounds were purchased as commercial preparations with a nominal purity of at least 95% from Sangon Biotech Company (Shanghai, China) and the pure compounds were analytical grade.

**Flat plate antimicrobial activity of pure VOCs.** The antifungal activity of selected individual pure VOCs on the mycelial growth of *A. alternata* was evaluated using the fumigation method (27). Sterilized Petri dishes (inner diameter 90 mm) were filled with 20 mL of PDA, and a mycelium plug (9-mm diameter) from the edge of 3-day-old *A. alternata* culture was placed on the PDA in the center of the petri

dishes. Individual compounds were pipetted onto sterile filter paper discs (10-mm diameter) to obtain the following range of concentrations: 0.1, 0.2, 0.3, 0.4, 0.5, and 0.6 $\mu L$ $mL^{-1}$. The PDA plate was then turned upside down (the side with fungal cake was on the top and the side with the volatile matter was on the bottom), sealed with double-deck parafilm, and incubated for 7 days at 25°C. Then, the lateral and longitudinal diameters of the colonies (perpendicular to each other) were measured with a ruler, and the average of the two as the final diameter of the colonies (6) was taken. Subsequently, the 50% inhibitory concentrations ($IC_{50}$s) was calculated (46), and each treatment was repeated three times. The inhibition of mycelial radial growth was calculated according to the following formula:

$$\text{Inhibition of mycelial growth } (\%) = [(S_1 - S_2)/S_1] \times 100$$

where, $S_1$ denotes the control's mean colony average diameters and $S_2$ denotes the treated groups' mean colony average diameters (49).

**Microscopic analysis.** Optical Microscopy (Olympus BX5X microscope) was used to assess the influence of VOCs (single ingredient) produced by *C. fimbriata* on the morphology of pathogen mycelia (50). The method described above of 2.4 was used after incubating cultures treated with different volatile compounds for 7 days (25°C), making slide samples, and morphology under an Olympus BX5X microscope (10 × 40 times, 20 $\mu$m). We then observed mycelial morphology, spores, and inclusions.

**Spore germination test of pure VOCs.** The spore germination of *A. alternata* was tested by the method of Li et al., and slightly improved (51). In brief, mix *100 $\mu L$* of spore suspension ($1.0 \times 10^6$ CFU $mL^{-1}$) with the compounds, and set the concentration of compounds to *50 $\mu L$ $mL^{-1}$, 100 $\mu L$ $mL^{-1}$*, and *200 $\mu L$* $mL^{-1}$. Then, the mixture was dripped onto the center of a concave slide and covered with a coverslip. The slide was incubated in a Petri dish with moist filter paper at 25°C for 12 h, and spore germination was tested under an optical microscope. The germination was taken into account when the germ tube length exceeded half of the conidia's length. Five view fields were examined for each treatment to calculate the germination rate. Each treatment consisted of three replicates, and no compound was added as a control.

***In vivo* antifungal assay test of pure VOCs.** Based on the results of in plate antibacterial testing, four kinds of pure components comprising the VOCs were selected to test inhibition of pathogen growth *in vivo*. Individual VOCs with potent *in vivo* antifungal activity against *A. alternata* were evaluated for their effects on fungal development on artificially inoculated cherry tomatoes. We chose fresh mature tomato fruit that was bright in red color and uniform in size. The weight of each cherry tomato fruit was about 30 ± 5 g, and then washed and the surface disinfected with 75% alcohol (30 s), and dried naturally (25°C). The *in vivo* assay was based on Zhao et al.' s method, with some modifications (37). Each tomato fruit was punctured at a point with a puncher (5 mm); pathogen starter cultures were grown on PDA for 3 days, and 5-mm punches were used to make fungus cakes that closely fit the wound. The inoculated fruit was arranged in moistened 4-L sealed containers, with 10 fruit per treatment. Filter paper discs (12.0-cm diameter) were pasted on container lids, and individual volatile compounds were added to discs to obtain the concentrations of 0.0125, 0.025, 0.05, 0.1, and 0.*2 $\mu L$* $mL^{-1}$ of container volume. The inoculated fruit was placed in boxes without the synthetic compound as the control. The boxes containing inoculated fruit were incubated for 7 days at 25°C. The proportion of rot lesions was calculated according to the following formula:

$$\text{lesion diameter } (mm) = [(T_1 - T_2)/T_1] \times 100$$

where, $T_1$ is the mean average proportion of the control of rot lesion (mm), and $T_2$ is the mean the average proportion of rot lesion (mm) of the treated groups.

**Statistical analysis.** The data were analysis calculated using Excel 2010 (Microsoft), and the means and standard deviations were calculated in all tests. Statistically, significant differences between different treatments were assessed using Duncan's multiple range test at $P < 0.05$ with SPSS Version 23.0 software. All data are expressed as mean ± standard deviation (X ± SD).

**Data availability.** The sequence information of the two *C. fimbriata* strains used in this study have been uploaded to the NCBI database. The gene accession numbers are MH535912 (WJSK-1), KY580883 (Mby).

## ACKNOWLEDGMENTS

This research was supported by the National Natural Science Foundation of China (31860522) and the National Key Research and Development Program of China (2019YFD1002004).

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
