## [Reviewer comments · Microbiology Spectrum]

Microbiology Spectrum

Antifungal activity of volatile components from *Ceratocystis fimbriata* and its potential biocontrol mechanism on *Alternaria alternata* in postharvest cherry tomato fruit 搜索复制

Shijun Xing, Yating Gao, Xue Li, Huan Ren, Yang Gao, Hui Yang, Yanmei Liu, Shuqi He, and Qiong Huang

Corresponding Author(s): Qiong Huang, Yunnan Agricultural University

Review Timeline:

Submission Date:	July 16, 2022
Editorial Decision:	August 25, 2022
Revision Received:	October 31, 2022
Editorial Decision:	November 28, 2022
Revision Received:	December 6, 2022
Accepted:	December 8, 2022

Editor: Gustavo Goldman

Reviewer(s): Disclosure of reviewer identity is with reference to reviewer comments included in decision letter(s). The following individuals involved in review of your submission have agreed to reveal their identity: Mengyao Niu (Reviewer #2)

Transaction Report:

DOI: <https://doi.org/10.1128/spectrum.02713-22>

August 25, 2022

Prof. Qiong Huang
Yunnan Agricultural University
China
Kunming
China

Re: Spectrum02713-22 (Antifungal activity of volatile components from *Ceratocystis fimbriata* and its potential biocontrol mechanism on *Alternaria alternata* in postharvest cherry tomato fruit搜索复制)

Dear Prof. Qiong Huang:

Your manuscript has been reviewed by two reviewers who provided some suggestions for improving it. Please, submit a revised version together with a rebuttal letter addressing point-by-point raised by each reviewer.

Link Not Available

Sincerely,

Gustavo Goldman

Journals Department
Reviewer comments:

Reviewer #1 (Comments for the Author):

Post harvest damage is a major problem for agricultural produce, and new ways to control the pathogens responsible are needed. Here, the authors found that volatiles from *Ceratocystis fimbriata* inhibit growth, conidiation and spore germination of *Alternaria alternata*, and cause abnormal mycelial morphology. Detailed characterization of the VOCs led to four candidates for further study as individual volatile antifungal metabolites.

The VOCs analyses look technically sound and the inhibitory effects are clear.

The rationale for the research is not fully developed in the manuscript. *Ceratocystis fimbriata* is, itself, a soil-borne Sordariomycete plant pathogen. In the introduction, it would be relevant to explain whether *C. fimbriata* and *A. alternata* coexist in the niche, are they antagonists, what was the basis for the idea that *C. fimbriata* volatiles would be good suppressors of *A. alternata* disease? The compounds studied are anti-microbial, but what is the significance of another fungus as their source? From the agricultural/post-harvest point view, how do these compounds differ, having a biological source, from chemical treatments? Biocontrol potential would imply co-culture, rather than only exposure of the pathogen (*Alternaria*) to the compounds produced by *C. fimbriata*.

other notes and minor issues to be addressed:

lines 145-146 - the accession numbers are for ITS sequences in Genbank, not the strains. Please rewrite, and also give the source of the strains.

line 148 - what is "activated" (retrieved from frozen storage?)

line 151 - something is missing, for example "Our aim was to study..."

line 168 and elsewhere - "antibacterial" better antimicrobial (to include fungi)

line 182-184 - this sentence is unclear - is "control effect" a control treatment, or biocontrol = inhibition of the pathogen?

line 200 "cross method" and elsewhere "cross cross" this doesn't seem to be standard terminology; could simply state how the diameter was measured, "using ImageJ on digital images" or "with a ruler" etc

line 219 - I don't think this refers to a concentration gradient, rather a series of concentrations tested. Here and elsewhere, please standardize units: 20 microliters/L or microliters/mL (not total microliters)

line 239 - "activated" - better: starter cultures were grown on [what medium] for three days

line 244 - please standardize here and elsewhere, microL/L or microL/mL

line 278 and within Table 1 - hydncarpates - please check, could be hydrocarbons?

line 309 delete "its"

line 324 - The control, in contrast, sporulated (Figure 3).
also - Figure 3 - scale bars - bar and font are too small

line 342 - as elsewhere - please standardize units

line 351 - should be Table 4

line 390 - scientific literature - and, Table 5 is a very useful review in its own right, however it is hardly cited in the text

line 433 - how does the discussion of this compound fit into the study? I don't see it anywhere else, is it produced by *C. fimbriata*?

line 440 "the same single component" - which one(s) - obviously the four main candidates, but please spell this out for the reader - are they found in many other species - and refer to Table 5

line 458 - mode of action - this is a bit of an overstatement

Reviewer #2 (Comments for the Author):

Please see attached for comments.

Staff Comments:

Preparing Revision Guidelines

Please return the manuscript within 60 days; if you cannot complete the modification within this time period, please contact me. If you do not wish to modify the manuscript and prefer to submit it to another journal, please notify me of your decision immediately so that the manuscript may be formally withdrawn from consideration by Microbiology Spectrum.

Spectrum02713-22_Reviewer Comments

Summary:

In current study, the authors characterized the antifungal effects of *C.fimbriata* against the plant pathogen *A.alternata* through inhibiting spore germination and altering hyphal growth. The effects of VOCs are clearly demonstrated and composition of VOCs in the culture headspace was surveyed. Four VOCs were studied in more depth for their effects against *A.alternata* growth and pathogenicity. Overall the study was well-designed and represented with clear text messages. However, there are a few controls missing in the study that need to be addressed.

Introduction:

Line 19/20: grammatical error

Line 60: Double check this citation for accuracy

Line 75: requires citation

Line 84: This study only shows bacterial VOCs antagonizing fungi

Materials&Method:

Line 160: check this sentence

Results:

Figure 1: Did *A.alternata* have any effect on the growth of WJSK and Mby strains? Please show images of the co-culture and the antifungal strains.

Table 1: Legend is missing. What does the % area refer to? Is this representative of the relative amounts of VOCs in the headspace of these fungal cultures? The VOCs analysis samples were acquired from headspace of the co-culture, do you have single-culture controls? Did co-culture change VOC profiles? Without comparison to a control, the VOCs listed here can be released from either *A. alternata* or *C.fimbriata*.

Line 292: According to table 1. Most of the 4, except for the isoamyl acetate are present in quite small quantity in the fungal culture headspace. How are these 4 synthetic VOCs chosen? How do IC50 values compare to the native levels?

Figure 3: Cool findings. But what does *A.alternata* hyphae look like when exposed to total VOCs? This is important control, suggesting the altered hyphal morphology is relevant to exposure to *C.fimbriata* VOCs.

Figure5: Like experiment for figure 3, a co-culture and single-culture headspace control should be included to examine if *C.fimbriata* releases VOCs that control the pathogenic growth of *A.alternata* on tomatoes.

Discussion:

Reply to the comments of reviewers

<Microbiology Spectrum>

<Antifungal activity of volatile components from *Ceratocystis fimbriata* and its potential biocontrol effect on *Alternaria alternata* in postharvest cherry tomato fruit>

< Paper #Spectrum02713-22>

Dear reviewers,

We are grateful to the reviewers for giving us the comments our manuscript. We checked the manuscript again and thought deeply about the issues raised by the reviewers, and the detailed corrections are listed point by point below:

Reviewer 1

1. The rationale for the research is not fully developed in the manuscript. *Ceratocystis fimbriata* is, itself, a soil-borne Sordariomycete plant pathogen. In the introduction, it would be relevant to explain whether *C. fimbriata* and *A. alternata* coexist in the niche, are they antagonists, what was the basis for the idea that *C. fimbriata* volatiles would be good suppressors of *A. alternata* disease?

Reply: Thanks to the reviewer for the questions. As we know that biocontrol agents (BCAs) as a biofumigant is a specific application in biological control since they are not in direct contact with the pathogen. microbial VOCs easily volatilize and degrade at room temperature and do not easily remain on the food surface. VOCs applies not only to different storage stages of various products but also to products that are easily damaged and cannot be treated with liquid fungicides, such as vegetable, fruit, etc. So, it is a promising application direction for VOCs to prevent postharvest decay as

biological fumigation. Although *C. fimbriata* strains are soil-borne ascomycete fungus, previously reported that VOCs from *C. fimbriata* had a strong inhibitory effect on the test fungi including *Botrytis cinerea*, *Monilinia fructicola*, *Fusarium sp.*, *Penicillium sp.*, *Rhizoctonia solani*, but the inhibition mechanism of single VOCs against fungus had not been investigated. Our aim is to explore their bacteriostatic components. In addition, we rewrote the research background, especially in **page 5, line 95-102 of marked up manuscript.**

2. The compounds studied are anti-microbial, but what is the significance of another fungus as their source?

Reply: Maybe we can use those anti-microbial compounds as green fumigants (such as 2-Phenylethanol etc.) replace chemical fumigants(toxic) in the future.

3. From the agricultural/post-harvest point view, how do these compounds differ, having a biological source, from chemical treatments?

Reply:

1) Same as 2

2) Maybe we can collect VOCs through *C. fimbriata* strains fermentation.

4. Biocontrol potential would imply co-culture, rather than only exposure of the pathogen (*Alternaria*) to the compounds produced by *C. fimbriata*.

Reply : Yes, co-culture method fit for non-violate anti-microorganism screening. We use to exposure of the pathogen (*Alternaria*) to VOCs produced by *C. fimbriata* because only the VOCs produced by *C. fimbriata* not itself has the antifungal activity.

Which method is common used in those production VOCs anti-microorganism

screening.

5

5.1 lines 145-146 - the accession numbers are for ITS sequences in GenBank, not the strains. Please rewrite, and also give the source of the strains.

Reply: Thanks for the kind reminder. The information of the two strains (WJSK-1 and Mby) has been added to the manuscript. And WJSK-1 was isolated from *Lactuca sativa* var. *angustana* Irish (Gene bank number MH535912), the Mby was isolated from *Musa basjoo* Siebold (Gene bank number KY580883). Please see the following picture. (Marked up manuscript page 12, line 232-235).

GenBank Send to: ▾

Ceratocystis fimbriata isolate WJSK-1 small subunit ribosomal RNA gene, partial sequence; internal transcribed spacer 1, 5.8S ribosomal RNA gene, and internal transcribed spacer 2, complete sequence; and large subunit ribosomal RNA gene, partial sequence

GenBank: MH535912.1
FASTA Graphics

Go to: ☺

LOCUS MH535912 750 bp DNA linear PLN 26-NOV-2018
DEFINITION Ceratocystis fimbriata isolate WJSK-1 small subunit ribosomal RNA gene, partial sequence; internal transcribed spacer 1, 5.8S ribosomal RNA gene, and internal transcribed spacer 2, complete sequence; and large subunit ribosomal RNA gene, partial sequence.
ACCESSION MH535912
VERSION MH535912.1
KEYWORDS .
SOURCE Ceratocystis fimbriata
ORGANISM Ceratocystis fimbriata
Eukaryota; Fungi; Dikarya; Ascomycota; Pezizomycotina;
Sordariomycetes; Hypocreomycetidae; Microascales;
Ceratocystidaceae; Ceratocystis.
REFERENCE 1 (bases 1 to 750)
AUTHORS Li, X., Sun, Y. X. and Huang, Q.
TITLE Ceratocystis fimbriata isolated from different hosts in China
JOURNAL Unpublished
REFERENCE 2 (bases 1 to 750)
AUTHORS Li, X., Sun, Y. X. and Huang, Q.
TITLE Direct Submission
JOURNAL Submitted (24-JUN-2018) College of Plant Protection, Yunnan Agricultural University, HeiJin Road, Kunming, Yunnan 650201, China
FEATURES Location/Qualifiers

WJSK-1

GenBank Send to: ▾

Ceratocystis fimbriata isolate Mby internal transcribed spacer 1, partial sequence; 5.8S ribosomal RNA gene and internal transcribed spacer 2, complete sequence; and large subunit ribosomal RNA gene, partial sequence

GenBank: KY580883.1
FASTA Graphics

Go to: ☺

LOCUS KY580883 537 bp DNA linear PLN 12-JUN-2017
 DEFINITION Ceratocystis fimbriata isolate Mby internal transcribed spacer 1, partial sequence; 5.8S ribosomal RNA gene and internal transcribed spacer 2, complete sequence; and large subunit ribosomal RNA gene, partial sequence.
 ACCESSION KY580883
 VERSION KY580883.1
 KEYWORDS
 SOURCE Ceratocystis fimbriata
 ORGANISM Ceratocystis fimbriata
 Eukaryota; Fungi; Dikarya; Ascomycota; Pezizomycotina;
 Sordariomycetes; Hypocreomycetidae; Microascales;
 Ceratocystidaceae; Ceratocystis.
 REFERENCE 1 (bases 1 to 537)
 AUTHORS Zhang, Y., Sun, Y.X. and Huang, Q.
 TITLE Ceratocystis fimbriata isolated from different hosts in China
 JOURNAL Unpublished
 REFERENCE 2 (bases 1 to 537)
 AUTHORS Zhang, Y., Sun, Y.X. and Huang, Q.
 TITLE Direct Submission
 JOURNAL Submitted (06-FEB-2017) College of Plant Protection, Yunnan Agricultural University, Heijian Road, Kunming, Yunnan 650201, China
 FEATURES
 Location/Qualifiers
 source
 1..537
 /organism="Ceratocystis fimbriata"
 /mol_type="genomic DNA"

Mby

5.2 line 148 - what is "activated" (retrieved from frozen storage?)

Reply: Yes, we retrieved from frozen storage.

"activated" is replaced by " purified "

Correction: purified three times on PDA plates in advance before use (Marked up manuscript **page 13, line236**).

5.3 line 151 - something is missing, for example "Our aim was to study..."

Reply:

Original: To study.....

Correction: Our aim was to study... (Marked up manuscript **page 12, line238**).

5.4 line 168 and elsewhere - "antibacterial" better antimicrobial (to include fungi)

Reply:

Original: "antibacterial"

Correction: " antimicrobial "(Marked up manuscript **page 13, line251**).

5.5 line 182-184 - this sentence is unclear - is "control effect" a control treatment, or biocontrol = inhibition of the pathogen?

Reply:

Original: 21 compounds were selected from Table 1 and tested for the antifungal effect of *A. alternata* plates and the control effect of post-harvest tomatoes by artificial inoculation of *A. alternata*.

Correction: 21 compounds were selected from Table 1 and the antifungal activity of single volatile compounds were determined both in vitro and in vivo for pathogen *A. alternata*. (Marked up manuscript **page 13, line261-263**).

5.6 line 200 "cross method" and elsewhere "cross cross" this doesn't seem to be standard terminology; could simply state how the diameter was measured, "using ImageJ on digital images" or "with a ruler" etc4

Reply:

Original: The colony diameter was measured using the cross method.

Correction: Then, measure the lateral and longitudinal diameters of the colonies (perpendicular to each other) with a ruler, and take the average of the two as the final diameter of the colonies. (Marked up manuscript **page 14, line274-276**).

5.7 line 219 - I don't think this refers to a concentration gradient, rather a series of concentrations tested. Here and elsewhere, please standardize units: 20 microliters/L or microliters/mL (not total microliters)

Reply:

Original: 5 μL , 10 μL , and 20 μL

Correction: 50 μLmL^{-1} , 100 μLmL^{-1} , and 200 μLmL^{-1} (Marked up manuscript **page 15, line290**).

5.8 line 239 - "activated" - better: starter cultures were grown on [what medium] for three days

Reply:

Original: pathogens were activated for 3 d.

Correction: pathogen starter cultures were grown on PDA for three days. (Marked up manuscript page 15, line303-304).

5.9 line 244 - please standardize here and elsewhere, microL/L or microL/mL

Reply:

same as 5.7 (Marked up manuscript page 15, line307-308).

5.10 line 278 and within Table 1 - hydnocarpates - please check, could be hydrocarbons?

Reply:

Original: hydnocarpates

Correction: Aromatic hydrocarbons (Marked up manuscript page 6, line116-117).

5.11 line 309 delete "its"

Reply:

Already deleted.

5.12 line 324 - The control, in contrast, sporulated (Figure 3) also - Figure 3 - scale bars - bar and font are too small.

Reply:

Original Fig.3:

Correction Fig.3: Please see the picture below (Marked up manuscript page7, line139-140)

Figure 3 Effect of VOCs from *C. fimbriata* on the hyphal morphology of *A. alternata*. After exposure to the four pure volatiles and the total VOCs of *C. fimbriata*, and incubation on potato dextrose agar (PDA) at 25°C for 7 d, microscopic morphological changes of *A. alternata*, (A) benzaldehyde, (B) nonanal, (C) 2-Phenylethanol, (D) isoamyl acetate, (E) strain WJSK-1, (F) strain Mby.

5.13 line 342 - as elsewhere - please standardize units

Reply: same as 5.7 (Marked up manuscript page 25, line517).

5.14 line 351 - should be Table 4

Reply:

Yes, thanks for your reminder, we corrected it. (Marked up manuscript page 8, line151).

5.15 line 390 - scientific literature - and, Table 5 is a very useful review in its own

right, however it is hardly cited in the text

Reply:

We rewrote the discussion, please see 1-2 paragraph as follow: (Marked up manuscript **page 9 line175-page 11 line208**).

DISCUSSION

Postharvest decay is one of the main factors that determine losses and eating quality of vegetables and fruit. One of the most harmful fungi, *A. alternata*, can cause postharvest rot and black rot in tomatoes after harvest (39). As the disadvantages (toxicity and generation of pathogen resistance) of chemically synthesized fungicides gradually emerge, microbial metabolites will become one of the most likely replacements. Microorganisms produce a variety of VOCs, which are gas-phase, carbon-based compounds that can diffuse through the atmosphere and soils due to their small size(14). Therefore, the use of volatiles naturally produced by microorganisms for "biological fumigation" has become one of the research hotspots(15, 30). The inhibiting the growth of plant pathogens of volatile organic compounds (VOCs) derived from microorganisms have been reported in the scientific, and they have gained widespread attention from researchers(13). According to previous research, that volatile-producing fungus, bacteria and plant have a positive effect on biological fumigation for the prevention and control of fruit postharvest diseases. Fungi such as *Muscodor albus*, *Penicillium* (*Penicillium expansum*, *Penicillium italicum*), *C. fimbriata*, endophytic fungi(27, 40-42), bacteria such as *Bacillus velezensis*, *Bacillus siamensis*, *Pseudomonas fluorescens*, *Corallocooccus.sp*(30, 36, 37), and the plant *Origanum vulgare*(31), their volatiles can

effectively inhibit the growth of many pathogens, including *Botrytis cinerea*, *Colletotrichum*, *Geotrichum*, *Monilinia*, *Penicillium*, *Rhizopus*, *Macrophomina phaseolina*, *Monilinia fructicola* and *Fusarium*(Table 5). Like our study, the VOCs produced by *Aureobasidium pullulans* also inhibit the growth of *A. alternata* (Table 5). In addition, some plant products have been recommended as safe alternatives to synthetic antimicrobials, eg., Essential oils(9, 31, 43).

Previously reported that VOCs from *C. fimbriata* had a strong inhibitory effect on the test fungi, oomycetes, and bacteria, but the inhibition mechanism of single VOCs against *A. alternata* had not been investigated(27). Except 2-Phenylethanol from *Aureobasidium pullulans* inhibited *A. alternata* (Table 5), this study found that other three single volatile components from *C. fimbriata* can inhibit the growth of *A. alternata*. i.e. benzaldehyde, nonanal and isoamyl acetate. In addition, Phenylethyl alcohol from endophytic fungi inhibited *Fusarium*, benzaldehyde and 1-Phenylethanol and nonanal from *B. velezensis* inhibited *B. cinerea* and *Penicillium* (Table 5). Based on Table 5, benzaldehyde, nonanal and 2-Phenylethanol were proved antifungal activity, and isoamyl acetate was reported to be effective in inhibiting *Curvularia lunata*(44). Secondly, 2-Phenylethanol is not only a common ingredient in flavors (20), but also has excellent antifungal activity(45).

5.16 line 433 - how does the discussion of this compound fit into the study? I don't see it anywhere else, is it produced by *C. fimbriata*?

Reply:

Already deleted in the manuscript.

5.17 line 440 "the same single component" - which one(s) - obviously the four main

candidates, but please spell this out for the reader - are they found in many other species - and refer to Table 5

Reply:

Original: The single component from fungi and bacteria has been shown to have antifungal effects against some major pathogens (*B. cinerea*, *A. alternata*, *Penicillium*, *Fusarium*) (Table 5), and the same single component was detected in the *C. fimbriata* strains. In this study, among the four volatiles tested, in addition to isoamyl acetate, benzaldehyde, nonanal, and 2-Phenylethanol are more competitive in antifungal activity for *A. alternata*.

Correction: Some single components produced by fungi and bacteria have been shown to have antifungal effects against some major pathogens (*B. cinerea*, *A. alternata*, *Penicillium*, *Fusarium*) (Table 5), among the single components that have been reported to have antifungal activity in a part listed in Table 5, three components, namely benzaldehyde, nonanal, and 2-Phenylethanol, were also detected in the VOCs of the *C. fimbriata* strains, and their antifungal activities were confirmed again in our study.

We rewrote the discussion, please see 2 paragraphs in 5.15: (Marked up manuscript page 9-10, line182-192 and page 11, line210-217).

5.18 line 458 - mode of action - this is a bit of an overstatement

Reply:

Thanks, we changed it to “antifungal mechanism” (Marked up manuscript page 12, line246).

Thanks again to the reviewer for the questions for us.

Reviewer 2

1. Introduction:

Line 19/20: grammatical error

Reply:

Original: then used HS-SPME-GC-MS technology to analyze the volatiles produced by two strains *C. fimbriata* (WJSK, Mby),

Correction: then HS-SPME-GC-MS technology was used to analyze the volatiles produced by two strains of *C. fimbriata* (WJSK-1, Mby), (Marked up manuscript page1, line19-20).

Line 60: Double check this citation for accuracy.

Reply:

Original: Post-harvest rot caused by these fungi can cause serious economic losses and limits the duration of storage and shelf-life of the fruit and vegetables, and reduction of the market value of fruit due to the mycotoxin contamination.

Correction: The occurrence of post-harvest diseases is one of the main factors that cause economic losses during post-harvest storage and transportation, which not only affects the storage time and shelf life of fruits and vegetables, but also leads to a decline in market value. (Marked up manuscript page3, line54-56).

Line 75: requires citation

Reply: Now, two references are cited.

Original: In addition to chemical synthetic fungicides, microorganisms that produce volatile organic compounds (VOCs) have recently received a lot of attention, VOCs are considered promising alternatives for post-harvest fungi management.

Correction: In addition to chemical synthetic fungicides, microorganisms that produce volatile organic compounds (VOCs) have recently received a lot of attention, VOCs are considered promising alternatives for post-harvest fungi management(14, 15) (Marked up manuscript page4, line66-69).

Line 84: This study only shows bacterial VOCs antagonizing fungi

Reply: Yes, because we focus on postharvest fungal pathogens, especially *A. alternata*.

Original: *Aureobasidium pullulans*, *Metschnikowia fructicola*, and *Bacillus amyloliquefaciens*, etc, can produce metabolites that antagonize much of pathogens and be used to control plant diseases.

Correction: *Aureobasidium pullulans*, *Metschnikowia fructicola*, and *Bacillus amyloliquefaciens*, etc, can produce VOCs metabolites that can control the growth of postharvest fungal pathogens on fruit (Marked up manuscript page4, line70-71)

2. Materials &Method:

Line 160: check this sentence

Reply:

Original: The colony diameter *A. alternata* the control and treatment were measured, and observe the colony morphology.

Correction: The colony diameter of *A. alternata* in the control and treatment groups were measured, and observe the colony morphology. (Marked up manuscript page12, line245-246).

3. Results:

3.1 Figure 1: Did *A. alternata* have any effect on the growth of WJSK-1 and Mby strains? Please show images of the co-culture and the antifungal strains.

Reply: We only have that VOCs produced by WJSK-1 and Mby strains inhibited *A. alternata*. The images of the co-culture please see Fig.1B, the antifungal strains colony WJSK-1 and Mby see Fig.1A---we added in Fig.1.

Fig. 1A

Fig. 1B

In short, *C. fimbriata* strains were grown on PDA for 7 d, and the lid of each Petri dish was removed, and the plate containing PDA inoculated with tested fungi was replaced, and the two plates were then sealed together with Parafilm, the dual-culture plates were then incubated at an ambient temperature (25°C) for 7 d, PDA plate not inoculated with *C. fimbriata* strain as a control (please see the marked up manuscript page12 line241-245).

3.2 Table 1: Legend is missing. (1) What does the % area refer to? Is this representative of the relative amounts of VOCs in the headspace of these fungal cultures? (2) The VOCs analysis samples were acquired from headspace of the co-culture, do you have single-culture controls? (3) Did co-culture change VOC profiles? Without comparison to a control, the VOCs listed here can be released from either *A. alternata* or *C. fimbriata*.

Reply: (1) Yes (Marked up manuscript page29, line544-545). (2) The VOCs analysis

samples were acquired only from headspace of the *C. fimbriata* culture because *A. alternata* culture cannot produce VOCs. (3) If co-culture change VOC profiles is not clear, the VOCs listed here can be released only from *C. fimbriata*.

3.3 Line 292: According to table 1. Most of the 4, except for the isoamyl acetate are present in quite small quantity in the fungal culture headspace. How are these 4 synthetic VOCs chosen? How do IC₅₀ values compare to the native levels?

Reply: In the early stage, we did a lot of screening work. Among the single components detected, we selected 21 compounds that were easily obtained as the screening objects, rather than only selecting the components with high content for research. Finally, we found that the inhibitory effect of the four single components on the postharvest pathogen *A. alternata* was better than that of other components. It is speculated that the four components the key factor for the *C. fimbriata* strain to inhibit the growth of *A. alternata*. Although 3 of the 4 components, except for the isoamyl acetate are present in quite small quantity in the fungal culture headspace, maybe inhibition *A. alternata* as synthetic function with other many compounds together. In addition, benzaldehyde and 2-Phenylethanol are proved to have antifungal activity before. Our research aims to explore whether it is feasible for these components to become fumigants to delay the onset of vegetables or fruits in the future, so these compounds have become our focus. Next, we will increase the workload in this area. As for the IC₅₀ value compared with the native, the inhibitory effect of the pure product is stronger than that of the native, which may be due to the fact that the native contains other non-antifungal components (such as growth-promoting compounds, of which 2-Pentyl-Furan has been proven to play a key role in promoting plant growth), which may result in different function.

3.4 Figure 3: Cool findings. But what does *A. alternata* hyphae look like when

exposed to total VOCs? This is important control, suggesting the altered hyphal morphology is relevant to exposure to *C. fimbriata* VOCs.

Reply: Yes, the same antifungal mechanism whether total VOCs or single compound. For total VOCs, we added this part of the experiment, and the morphological changes and sporulation of *A. alternata* hyphae exposed to the total VOCs of *C. fimbriata* in Figure 3E and 3F (Please see the following picture).

Figure 3 Effect of VOCs from *C. fimbriata* on the hyphal morphology of *A. alternata*. After exposure to the four pure volatiles and the total VOCs of *C. fimbriata*, and incubation on potato dextrose agar (PDA) at 25°C for 7 d, microscopic morphological changes of *A. alternata*, (A) benzaldehyde, (B) nonanal, (C) 2-Phenylethanol, (D) isoamyl acetate, (E) strain WJSK-1, (F) strain Mby.

3.5 Figure5: Like experiment for Figure 3, a co-culture and single-culture headspace control should be included to examine if *C. fimbriata* releases VOCs that control the pathogenic growth of *A. alternata* on tomatoes.

Reply: Same as Figure 3, we redesigned and carried out this part of the experiment, inoculating tomato fruit with *A. alternata*, and then with the *C. fimbriata* strains co-culture, the diameter of the lesions of tomato was reduced compared with the control, confirming that the volatiles produced by the *C. fimbriata* strains can control *A. alternata*. growing on tomato fruit, the results are shown in Figure 5 (Please see the following picture).

Figure 5 Antifungal effect of *C. fimbriata* volatiles and four single VOCs in vivo. Inhibitory effect activities of *C. fimbriata* volatiles and four single compounds against *A. alternata* on cherry tomatoes after 7 d of storage at 25°C. (C) Treatment with *C. fimbriata* volatiles. Treatment with (D-a) benzaldehyde; (D-b) nonanal; (D-c) 2-Phenylethanol; (D-d) isoamyl acetate; (A and B) lesion diameter of different tomato fruit treated with *C. fimbriata* volatiles, and four single components. Different letters indicate differences ($p < 0.05$) in the same storage time and the same fungus from different treatments. Bars represent mean values \pm standard deviation of 10 fruit.

4. Discussion:

Thanks again to the reviewer for the questions for us.

November 28, 2022

Prof. Qiong Huang
Yunnan Agricultural University
China
Kunming
China

Re: Spectrum02713-22R1 (Antifungal activity of volatile components from *Ceratocystis fimbriata* and its potential biocontrol mechanism on *Alternaria alternata* in postharvest cherry tomato fruit搜索复制)

Dear Prof. Qiong Huang:

One of the reviewers has provided some observations. Please, submit a revised version together with a rebuttal letter addressing point-by-point raised by this reviewer.

Link Not Available

Sincerely,

Gustavo Goldman

Journals Department
Reviewer comments:

Reviewer #1 (Comments for the Author):

It is shown here that *Ceratocystis fimbriata* volatiles are a source of compounds to inhibit *Alternaria alternata*, and hence can help reduce post-harvest damage. Post-harvest disease is a major problem for agricultural produce, as in the example studied here, black rot in cherry tomatoes. Detailed characterization of the VOCs led to four candidates for further study as individual volatile antifungal metabolites. In general, this revision has addressed my comments on the original version. In preparing the final version, I would just call the attention of the authors to two points that seem to have been missed in the revision:

238 Irish and Musa basjoo Sieboid, respectively, and preserved in our laboratory (GenBank accession 239 numbers were MH535912, KY580883)

please rewrite: "...in our laboratory." Ribosomal RNA gene sequences for the two strains can be found in Genbank, accession numbers MH535912 and KY580883.

line 295 and elsewhere: this is a concentration series, there is no gradient.
295 with the compound, set the compounds concentration gradient

"antibacterial" - better "antimicrobial" since this could be antifungal, as in the Alternaria example

overall, the manuscript would benefit from further editing for grammar & style. Examples:

in the Importance paragraph, several sentences start with "And"

line 43 - the sentence is incomplete

lines 96-103 - sentence structure could be improved

line 165 - food quality

line 172-175 could rewrite:

Inhibition of the growth of plant pathogens by volatile organic compounds (VOCs) derived from microorganisms has gained widespread attention (13).

Reviewer #2 (Comments for the Author):

Thanks for additional experimentation to add some of the controls in the study.

Staff Comments:

Preparing Revision Guidelines

Please return the manuscript within 60 days; if you cannot complete the modification within this time period, please contact me. If you do not wish to modify the manuscript and prefer to submit it to another journal, please notify me of your decision immediately so that the manuscript may be formally withdrawn from consideration by Microbiology Spectrum.

Reply to the comments of reviewers

<Microbiology Spectrum>

<Antifungal activity of volatile components from *Ceratocystis fimbriata* and its potential biocontrol effect on *Alternaria alternata* in postharvest cherry tomato fruit>

< Paper #Spectrum02713-22R1>

Dear reviewers,

We are grateful to the reviewers for giving us the comments our manuscript. We checked the manuscript again and thought deeply about the issues raised by the reviewers, and the detailed corrections are listed point by point below:

Reviewer 1

1. 238 Irish and Musa basjoo Sieboid, respectively, and preserved in our laboratory (GenBank accession

239 numbers were MH535912, KY580883)

please rewrite: "...in our laboratory." Ribosomal RNA gene sequences for the two strains can be found in Genbank, accession numbers MH535912 and KY580883.

Reply: Thanks to the reviewer for the questions.

Original: The *C. fimbriata* strains WJSK-1 and Mby were isolated and identified from *Lactuca sativa* var. *angustana* Irish and *Musa basjoo* Sieboid, respectively, and preserved in our laboratory (GenBank accession numbers were MH535912, KY580883).

Correction: The *C. fimbriata* strains WJSK-1 (isolated from *Lactuca sativa* var

angustana Irish) and Mby (isolated from *Musa basjoo Siebold*) were identified. Ribosomal RNA gene sequences for the two strains can be found in Genbank, accession numbers MH535912 and KY580883. (Marked up manuscript page 12, line 238-241)

2. line 295 and elsewhere: this is a concentration series, there is no gradient.

295 with the compound, set the compounds concentration gradient

Repley: Yes, Corrected

Original: In brief, mix 100 μL of spore suspension (1.0×10^6 CFU mL^{-1}) with the compound, set the compound concentrations gradient to 50 $\mu\text{L mL}^{-1}$, 100 $\mu\text{L mL}^{-1}$, and 200 $\mu\text{L mL}^{-1}$

Correction: In brief, mix 100 μL of spore suspension (1.0×10^6 CFU mL^{-1}) with the compound, and set the concentration of compounds to 50 $\mu\text{L mL}^{-1}$, 100 $\mu\text{L mL}^{-1}$, and 200 $\mu\text{L mL}^{-1}$.(Marked up manuscript page 15, line 295-296)

3. "antibacterial" - better "antimicrobial" since this could be antifungal, as in the Alternaria example

overall, the manuscript would benefit from further editing for grammar & style.

Examples: in the Importance paragraph, several sentences start with "And"

Reply: Corrected

Original: Based on the results of in plate **antimicrobial** testing, 4 kinds of pure components comprising the VOCs were selected to test inhibition of pathogen growth *in vivo*.

Correction: Based on the results of in plate **antibacterial** testing, 4 kinds of pure

components comprising the VOCs were selected to test inhibition of pathogen growth *in vivo*. (Marked up manuscript page 15, line 302)

Original: ~~And~~ most volatile organic compounds belong to five chemical groups: terpenes, fatty acid derivatives, benzene compounds, phenylpropanoid compounds, and amino acid derivatives (19).

Correction: In addition, most volatile organic compounds belong to five chemical groups: terpenes, fatty acid derivatives, benzene compounds, phenylpropanoid compounds, and amino acid derivatives (19). (Marked up manuscript page 4, line 76)

Original: ~~And~~ the four single compounds can effectively inhibit the spore germination of *A. alternata* (Fig. 4), the IC₅₀ of nonanal and benzaldehyde were 0.04 $\mu\text{L mL}^{-1}$ and 0.11 $\mu\text{L mL}^{-1}$.

Correction: Four of the single compounds can effectively inhibit the spore germination of *A. alternata* (Fig. 4), the IC₅₀ of nonanal and benzaldehyde were 0.04 $\mu\text{L mL}^{-1}$ and 0.11 $\mu\text{L mL}^{-1}$. (Marked up manuscript page 11, line 212)

4. line 43 - the sentence is incomplete

Reply : Yes, corrected

Original: Effect of single compound on *A. alternata* spore germination. The effect of the single compound on spore germination of the fungus is shown in Fig. 4.

Correction: The effects of different concentrations of the single compounds on the spore germination of *A. alternata* was tested *in vitro* (Fig. 4). (Marked up manuscript page 8, line 145-147)

5 lines 96-103 - sentence structure could be improved

Reply: Thanks for the kind reminder.

Original: The *C. fimbriata* strains are soil-borne ascomycete fungus, and it has also been considered a new aroma-producing strain, especially its volatile aroma is pleasant, similar to the fruity aroma (26, 27). Which has been studied for its ability to produce large quantities of VOCs, the GC-MS analysis of *C. fimbriata* VOCs revealed the presence of at least 28 VOCs from at least five classes of organic compounds, such as acids, alcohols, esters, and lipids (28), had a strong inhibitory effect on the test fungi including *Botrytis cinerea*, *Monilinia fructicola*, *Fusarium sp.*, *Penicillium sp.*, *Rhizoctonia solani*. But the inhibition mechanism of single VOCs against fungus had not been investigated (27, 29). Therefore, to explore their bacteriostatic components is meaningful.

Correction: *Ceratocystis fimbriata* is a typical soil-borne ascomycete that initially attracted attention due to its ability to cause disease in a broad range of economically important plants. Since then, with extended research, it has been shown to be a new aroma-producing fungus, capable of producing pleasant volatile aromas, similar to the fruity aroma (26, 27). Based on the widely used GC-MS technique in the analysis of microbial volatiles. It has been reported that *C. fimbriata* is capable of producing a large number of VOCs, analysis revealed the presence at least 28 single VOCs from at least category five of organic compounds, such as acids, alcohols, esters, and lipids (28). Of interest is that it shows strong antagonistic effects against a variety of pathogenic fungi, including *Botrytis cinerea*, *Monilinia fructicola*, *Fusarium sp.*, *Penicillium sp.*, and *Rhizoctonia solani*. But the inhibition mechanism of single VOCs against fungus has not been investigated (27, 29). Therefore, it is meaningful to explore the volatiles from *C. fimbriata* bacteriostatic components and to reveal

inhibition. (Marked up manuscript page 5, line 95-106)

6 line 165 - food quality

Reply: Thanks, corrected

Original: Postharvest decay is one of the main factors that determine losses and **eating quality** of vegetables and fruit.

Correction: Postharvest decay is one of the main factors that determine losses and **food quality** of vegetables and fruit. (Marked up manuscript page 9, line 168)

7 line 172-175 could rewrite:

Inhibition of the growth of plant pathogens by volatile organic compounds (VOCs) derived from microorganisms has gained widespread attention (13).

Reply: Corrected

Original: The inhibiting the growth of plant pathogens of volatile organic compounds (VOCs) derived from microorganisms have been reported in the scientific, and they have gained widespread attention from researchers (13)

Correction: Inhibition of the growth of plant pathogens by volatile organic compounds (VOCs) derived from microorganisms has gained widespread attention (13).(Marked up manuscript page 9, line 175-177)

We are again grateful to the reviewers for the work our manuscript.

Review 2

We are again grateful to the reviewers for the work our manuscript.

December 8, 2022

Prof. Qiong Huang
Yunnan Agricultural University
China
Kunming
China

Re: Spectrum02713-22R2 (Antifungal activity of volatile components from *Ceratocystis fimbriata* and its potential biocontrol mechanism on *Alternaria alternata* in postharvest cherry tomato fruit搜索复制)

Dear Prof. Qiong Huang:

Your manuscript has been accepted, and I am forwarding it to the ASM Journals Department for publication. You will be notified when your proofs are ready to be viewed.

Sincerely,

Gustavo Goldman
Editor, Microbiology Spectrum
